# Comparison of Morphological Sex Assessment and Genetic Sex Determination on Adult and Sub-Adult 17th–19th Century Skeletal Remains

**DOI:** 10.3390/genes14081561

**Published:** 2023-07-30

**Authors:** Tamara Leskovar, Teo Mlinšek, Tadej Počivavšek, Irena Zupanič Pajnič

**Affiliations:** 1Centre for Interdisciplinary Research in Archaeology, Department of Archaeology, Faculty of Arts, University of Ljubljana, 1000 Ljubljana, Slovenia; tamara.leskovar@ff.uni-lj.si; 2Institute of Forensic Medicine, Faculty of Medicine, University of Ljubljana, Korytkova 2, 1000 Ljubljana, Slovenia; tm5877@student.uni-lj.si (T.M.); tadej.pociva@gmail.com (T.P.)

**Keywords:** morphological sex assessment, genetic sex determination, method comparison, skeletal remains, osteobiography, bioarchaeology

## Abstract

**Highlights:**

Morphological methods for sex assessment achieved accuracy of 72% on non-adults and 97% on adults.Sex was assessed for 71% and determined for 94% of the individuals.Combining morphological and genetic method allowed all the individuals to be sexed.Applied morphological methods performed well on Slovenian post-medieval adults, while poorly on non-adults.

**Abstract:**

The first step in the analysis of human skeletal remains is the establishment of the biological profile of an individual. This includes sex assessment, which depends highly on the age of the individual and on the completeness and preservation state of the remains. Macroscopic methods only provide the assessment of sex, while for sex determination, molecular methods need to be included. However, poor preservation of the remains can make molecular methods impossible and only assessment can be performed. Presented research compares DNA-determined and morphologically assessed sex of adult and non-adult individuals buried in a modern-age cemetery (17th to late 19th century) in Ljubljana, Slovenia. The aim of the study was to assess the accuracy of commonly used macroscopic methods for sex assessment on a Slovenian post-medieval population. Results demonstrate that for adults, macroscopic methods employed are highly reliable and pelvic morphology, even the sciatic notch alone, is more reliable than skull. In non-adults, macroscopic methods are not as reliable as in adults, which agrees with previous research. This study shows how morphological and molecular methods can go hand in hand when building a biological profile of an individual. On their own, each methodology presented some individuals with undetermined sex, while together, sex of all the individuals was provided. Results confirm suitability of sex assessment based on skull and especially pelvic morphology in Slovenian post-medieval adults, while in the non-adult population molecular methods are advised.

## 1. Introduction

The first step in the analysis of human skeletal remains is the establishment of the biological profile of an individual. The basic biological profile is established using macroscopic morphological analysis and consists of the individual’s age, sex, stature, and ancestry. The first step in this analysis is a general assessment of age, which separates adults from non-adults. As many methods used for the establishment of the biological profile are sex-specific, the second and one of the most important steps in the analysis is sex assessment. Despite the apparent simplicity with limited options of male or female, sex assessment is one of the most difficult steps in the analysis. This is especially true when dealing with subadults, individuals with ambiguous sex characteristics, and/or poorly preserved remains [1]. Yet, sex assessment is a very powerful tool. In archaeological contexts, it is mostly used to better understand past human societies, demography, social stratification, gender roles, childhood developmental trajectory, burial practices, etc., while in forensic contexts it is one of the first steps in the identification of an unknown individual. Knowing the sex of an individual significantly helps with the identification process, as it narrows missing person searches by 50%. In case of commingled remains, it helps with the differentiation of the remains and determination of MNI (minimum number of individuals) [2].

Methodology for macroscopic sex assessment of a human skeleton consists of metric and non-metric methods. The first are based on the measurements of skeletal elements and the second on the observation of gross morphological traits, primarily of the skull and pelvis. Even though new methods are constantly appearing, and the range of skeletal elements used for sex assessment is expanding, the pelvis, especially the pubic bone, remains the most accurate. Somewhat less reliable are metric methods of the cranium and post-cranium, and methods based on the morphological traits of the cranium [3,4,5,6]. With numerous methods available, the process of sex assessment can vary significantly among physical anthropologists [7]. Furthermore, the preservation state can highly impact the possibility of performing sex assessment, as skeletal elements can be damaged or missing. Additionally, macroscopic methods for sex assessment of subadults are not accurate enough and are currently not advisable [8,9] as they need additional research [9].

For genetic sex identification, DNA markers on the Y and X chromosome are used and amplified in polymerase chain reaction (PCR). In routine forensic genetic investigations three sex-informative analyses are used and they can be implemented in sex assessment of aged skeletons. Amelogenin tests that are included in autosomal short tandem repeat (STR) typing kits amplify the targets on amelogenin genes’ intron 1 of the X and Y chromosome [10], and genetic markers on the Y chromosome are targeted in forensic quantitative PCR (qPCR) kits [11] and Y chromosome STR typing kits [12].

Macroscopic analysis can only offer an assessment of an individual’s sex and the possibility of misclassification is always present. Thus, molecular methods (analysis of peptides in enamel and DNA) are necessary for sex determination. On the other hand, poor preservation of the remains can make molecular methods impossible and only macroscopic assessment can be performed. Additionally, molecular-based sex determination also comes with various issues that complicate simple male/female dichotomy [13,14]. Furthermore, the Scientific Working Group for Forensic Anthropology’s document Sex Assessment [15] recommends an independent morphological analysis for sex assessment even if DNA analyses are conducted and sex can be determined by DNA.

DNA-based sex determination has significantly impacted the traditional workflow in the construction of biological profiles. Even though relatively rare and small, several studies comparing macroscopically assessed and DNA-determined sex exist [16,17,18,19,20,21,22]. As they present the pros and cons of each method and report inconsistency between the two, they clearly show that complementary work is the most efficient. Additionally, they indicate there is a potential for using more reliable DNA results to improve the accuracy of macroscopic sex assessment. They can target the problems which arise from the fact that macroscopic methods are mainly developed on geographically limited modern populations, yet they are applied to individuals living in different historic or prehistoric eras, and/or originating from different geographical areas. The accuracy of these methods applied on individuals from different populations (different period and/or location) is questionable as factors such as lifestyle, nutrition, growth and/or activity patterns, genetic admixture and disease can affect the sexual dimorphism between groups [23,24,25,26,27].

To the best of our knowledge, population-specific macroscopic methodologies for sex assessment of Slovenian populations, past or present, do not exist. As the most commonly employed macroscopic methods for sex assessment of adults and non-adults were developed on different populations and the population bias is a known fact, the aim of here presented research is to evaluate the reliability of these methods on a new age (17th to late 19th century) Slovenian population. To reach this aim, the macroscopically assessed and DNA-determined sex of individuals buried in a new age cemetery in Ljubljana were compared.

## 2. Materials and Methods

### 2.1. Description of the Ljubljana–Polje Archeological Site

Archaeological excavations north of the Church of the Virgin Mary in Polje, Ljubljana, Slovenia, which took place in August and September 2020, uncovered 216 inhumation graves from a modern-era cemetery. The cemetery was active between the early 16th and late 19th centuries [28,29] and can be divided into two phases. In the early phase, individual graves of adults predominated. Later, younger graves began to cut into the older ones or older graves were reused, likely due to lack of space. In this later phase, the cemetery was also separated into two parts. The western and central part were reserved for non-adults, while in the eastern part adults were buried. According to the remains of degraded wood and iron nails, the individuals were buried in wooden coffins. The majority of individuals was buried in the extended supine position with their hands at their sides or above the chest/pelvis area. They were buried clothed; some had small personal items such as rosaries, crosses, and pilgrim badges. Anthropological analyses were performed on all the excavated remains. Of all individuals, 29% were adults (>20 years old) and 71% were non-adults (<20 years old). Macroscopic sex assessment was possible for 75% of the adults, of whom 54% were females and 21% were males. Sex of non-adults was assessed only tentatively, as macroscopic methods are not highly reliable [30]. Sex of 31% of individuals was undetermined, 40% were possibly females and 29% were possibly males.

### 2.2. Samples

To establish biological profiles, anthropological analyses were performed on all the skeletal remains excavated from the Ljubljana–Polje archaeological site. Based on the completeness and preservation state of the skeletal remains, 116 individuals, comprising 32 adults and 84 non-adults, were selected for the research. After the anthropological analyses, petrous bones and, in nine cases without the petrous bone, femurs were collected for the ancient DNA analyses. The dense part of the petrous bone inside the otic capsule [31] and the diaphysis of femurs [12,32] were sampled. Using a sterilized diamond saw (Schick, Schemmerhofen, Germany), Pinhasi’s method [33] was used to detach the cochlea from the petrous bone. In femurs of non-adults below the age of 2, the epiphyses were cut away and the diaphysis was used for DNA extraction. In the rest, a piece of the diaphysis below the greater trochanter was sampled.

#### 2.2.1. Anthropological Analyses

Each individual skeleton was analyzed following the standards for recording human remains [34,35,36]. Based on the assessed sex, age at death, stature and pathological changes, biological profiles of individuals were established. Sex assessment of adult individuals was based on the morphological characteristics of the pelvis and the skull. For assessment of sex based on the pelvis, the subpubic region and/or the greater sciatic notch were examined following the Standards for Data Collection of Human Skeletal Remains [3,35,37]. For the assessment of sex based on the skull, the traits proposed by Acsádi and Nemeskéri [35,38] were used. The sex of non-adult individuals was assessed using the mandible and/or ilium as proposed by Loth and Henneberg [39] and Schutkowski [40]. In cases where the mentioned traits were not preserved, were too damaged for the examination or gave ambiguous results, sex was left undetermined. In cases where the traits of the skull and the traits of the pelvis presented different results, the assessment based on the pelvis was acknowledged. The methodological procedures for morphological sex assessment were chosen based on the completeness and preservation state of the remains, relative ease of use and frequency of use in the analyses of human skeletal remains.

#### 2.2.2. Genetic Analyses

DNA was extracted according to [41] and special measures [42,43] were followed to prevent contamination with contemporary DNA. For control of contamination, extraction-negative controls (ENC) were processed to monitor the purity of reagents and plastics used [44] and an elimination database was established including persons involved in analyzing DNA, in anthropological analyses, and in excavations. From these people, saliva was collected with sterile cotton swabs and DNA was extracted according to instructions provided by the manufacturer [45].

Three sex-informative tests were applied to perform genetic sex identification. The first one was a quantitative PCR (qPCR) test using the PowerQuant System (Promega)—qPCR Y-target test. The Y-chromosomal target included in the kit was used to detect the presence of male DNA. The qPCR test served also to determine the quantity (Auto target) and quality (Auto/Deg ratio) of DNA extracted from bones and ENCs. PowerQuant analyses were performed in duplicate following the technical manual [46] with the QuantStudio 5 Real-Time PCR system, the PowerQuant Analysis Tool (https://worldwide.promega.com/resources/tools/powerquant-analysis-tool/; accessed on 4 March 2023), and Quant-Studio Design and Analysis Software 1.5.1 (Applied Biosystems, AB, Foster City, CA, USA). The second sex-informative test was a Y-chromosomal short tandem repeat (STR) amplification test using the PowerPlex Y-23 kit (Promega)—Y-STR amplification test, and the third one was the amelogenin test included in the autosomal STR typing kit ESI 17 Fast (Promega). A PCR for Y-STR and autosomal STR typing was performed using 1 ng DNA whenever possible (in low-template samples and ENCs maximum DNA extract volume was used) following the recommendations of the manufacturer [47,48], using the Nexus MasterCycler (Eppendorf, Hamburg, Germany). Genetic profiles for the samples were acquired using the SeqStudio Genetic Analyzer for HID (Thermo Fisher Scientific, TFS, Waltham, MA, USA) combined with the WEN Internal Lane Standard 500 (Promega), SeqStudio Data Collection Software v 1.2.1 (TFS), and GeneMapper ID-X Software v 1.6 (TFS). All three sex-informative tests processed negative template controls and positive controls together with bone samples and ENCs.

Based on the qPCR results, low-template DNA samples (in PCR less than 100 pg DNA input was processed) that are prone to allelic drop-outs due to stochastic effects [49] were recognized and they were all analyzed using the Y-STR amplification test and amelogenin test (see Online Resource 1; seven samples labeled in blue). The Y-STR amplification test was also performed on all bone extracts (see Online Resource 1), in which the qPCR Y-target was detected (even if the measures were below the detection limit of the PowerQuant kit, which is 0.0005 ng [50] and was set in developmental validation for the PowerQuant System), and an additional amelogenin test was performed on those bone samples that produced partial Y-STR haplotypes consisting of 16 or fewer Y-STRs to confirm male sex of skeletons (see Online Resource 1). Some high-quantity samples with no Y qPCR target detected (see Online Resource 1) were also typed for Y-STRs to confirm no presence of male DNA. Altogether, 78 bone samples were typed for Y-STRs.

The absence of Y qPCR amplicons in bone extract that yielded more than 100 pg PCR input DNA indicates female sex of the skeleton. However, to additionally confirm female sex using an amelogenin sex test, some bones with different ages were selected (see Online Resource 1) and analyzed using the ESI 17 autosomal STR kit.

All ENC samples employed to monitor possible contamination with modern DNA were analyzed using the autosomal ESI 17 STR amplification kit, and Y-STR amplification was performed for ENC samples that produced PowerQuant Y-target amplicons (three out of 15 ENCs; see SM2, labeled in green).

In addition, all samples included in the elimination database were typed for autosomal STRs, and genetic profiles were compared to those obtained from bone samples to check the authenticity of isolated DNA and to exclude modern DNA contamination of the endogenous bone DNA through differentiation of genetic profiles of the bone compared to profiles in the elimination database. To increase the number of bone samples tested on autosomal STRs and compared to the elimination database, additional male skeletons were typed using the ESI 17 kit (see Online Resource 1).

#### 2.2.3. Statistical Analyses

Using Orange software [51], a Chi-square test was performed to see if the age of the individual impacted the matching of genetically determined and anthropologically assessed sex. First, comparisons were made between adults and non-adults. Second, non-adults were further grouped based on age (Table 1) and age groups compared.

## 3. Results

### 3.1. Morphological Sex

For 24 (29%) non-adult individuals, sex was undetermined, 43 (51%) were assessed as females and 17 (20%) as males. Seventeen (53%) adults were assessed as females and 15 (47%) as males (Table 2 and Appendix A). All non-adult individuals of undetermined sex were lacking the mandible and ilium, or the skeletal elements were too damaged for the assessment. In three cases, samples 10, 18 and 101, the sex assessment based on the mandible and the ilium did not match. As the ilium is more reliable for sex assessment, assessment based on the ilium was acknowledged. Similarly, the skull and ilium of two adults (samples 14 and 90) presented male and female characteristics and the assessment based on the ilium was acknowledged.

### 3.2. Genetic Sex

The Appendix A present bone sample characteristics and summarize the results for DNA quantity and quality (Auto, Deg, and Y-target—all values are presented as ng DNA/µL of extract, and Auto/Deg ratio) acquired by using the PowerQuant System (Promega). More than 0.5 pg of human DNA per μL of extract was detected in all skeletons except adult skeleton 1188 (see Online Resource 1). As expected for aDNA, the degradation index was high, reaching values up to 255, and in three cases the degradation index was undetermined because the Deg target could not be detected.

The Y PowerQuant target amplification product was identified in 74 samples (see Online Resource 1), and the quantity of male DNA was very low (mostly below the detection limit of the PowerQuant kit) in 25 of them (see Online Resource 1). All samples with detected PowerQuant Y-target amplicons were typed for Y-STRs, and only 51 produced Y-STR haplotypes using the PowerPlex Y-23 kit from Promega (see Online Resource 1). In 36 samples high quality Y-STR genetic profiles were acquired, and male sex was identified. In the other 15 samples, 16 or fewer STRs were amplified successfully and an additional amelogenin test was performed. In all samples but one, the amelogenin test confirmed male sex. After the Y-STR amplification test and in 15 samples, the additional amelogenin test analysis, male sex was confirmed for 49 skeletons (see Online Resource 1). For 22 samples with low Y-target measures and high measures of the Auto target and no Y-STR amplification success, female sex was identified (see Online Resource 1). It is possible to explain detection of a low quantity of the PowerQuant Y-target in those samples by observing that in three ENCs the qPCR Y-target also yielded a product (see SM2, labeled in green), but the ENC samples did not produce Y-STR haplotypes or amelogenin/STR profiles, which indicates that qPCR analysis has greater sensitivity for detecting minimal contamination issues. However, because the amounts of non-bone DNA were too low, this did not affect the detection of contamination alleles in Y-STR and autosomal STR profiles. Female sex was attributed to an additional 38 samples with high measures of Auto target and no Y-target detection, identifying altogether 60 skeletons as females.

In 109 skeletons sex was successfully determined using genetic methods (Table 2). Forty-three (51%) non-adults were females and 36 (43%) were males. Among adults, 17 (53%) were females and 13 (41%) were males. For seven individuals, two adults (6%) and five non-adults (6%), genetic sex was undetermined because of low quantity of extracted DNA. In low-template DNA samples, male sex could be confirmed only if all three tests were positive for the presence of Y-specific genetic markers. If not, it is also not possible to confirm female sex (see Supporting Information 1—two samples labeled in gray) because of the possibility of drop-outs due to stochastic effects [49].

Comparison of genetic profiles between the elimination database and bones was performed on all autosomal STR-typed bone samples that produced good-quality genetic profiles and no match was found. High degradation indexes, combined with pure negative controls also showed that the DNA obtained from ancient bones and the genetic profiles created were authentic to them.

### 3.3. Morphological and Genetic Sex Match

Twenty-nine individuals, 2 adults and 27 non-adults, with undetermined either morphological or genetic sex were excluded from comparisons (Appendix A). The morphologically assessed and genetically determined sex of 57 non-adult and 30 adult individuals was compared. A Chi-square test presented significant differences (X^2^ = 6.16, *p* = 0.013) in matching of genetic and anthropological sex of adult and non-adult individuals. Among adults, morphological sex was wrongly assessed in one case (3%). One female was assessed as male. In this case sex was only assessed based on the skull as pelvic bones were not preserved. Among non-adults, morphological sex was wrongly assessed in 16 cases (28%). In 11 cases (69%), sex was assessed as female instead of male, while in five cases (31%) as male instead of female. In four cases, sex was only assessed based on the mandible, and in four only based on the ilium. The Chi-square test presented no significant differences in the matching of genetic and anthropological sex of non-adults of different ages (X^2^ = 4.91, *p* = 0.178).

## 4. Discussion

The Chi-square test showed significant differences in the matching of genetic and anthropological sex between adults and non-adults, which was to be expected as morphological methods for the sex assessment of non-adults are less reliable [8,9]. The chosen methods for the sex assessment of the adult individuals performed well, as only one (3%) assessment was wrong. An adult female (sample 29) that was wrongly assessed as male lacked pelvic bones. The assessment was based only on the traits of the skull, which are known to be less reliable than the traits of the pelvis [30]. This is also seen in the cases presented here, as for the individuals with mismatched skull and pelvis morphological sex assessment, the genetic sex determination showed that pelvis-based assessment was the correct one. Additionally, the wrongly assessed individual (sample 29) was of advanced age, which could affect the morphology of the skull [30].

For adult individuals, stages 1 to 5 by Buikstra and Ubelaker [35] for the greater sciatic notch alone already presented accurate assessments. Twenty-three adult individuals lacked a pubic bone but had the greater sciatic notch preserved. The assessments for those individuals matched the genetic sex determination 100%, which is not the case with all populations [30,52,53,54]. This is important to know as the preservation state of the pubic bone, which is more reliable for sex assessment than the greater sciatic notch [3,30], is often poor or the bone is not preserved at all. For example, in the adult individuals included here, 72% lacked pubic bone.

It appears that for the studied population, the applied methodology for sex assessment of non-adults is not accurate enough for a reliable assessment, especially in forensic cases. The sex of 69% of individuals was assessed correctly, which is close to the accuracy obtained by other researchers [39,40,55,56]. The majority of wrongly assessed non-adults were males assessed as females. Vlak et al. [56] suggested that the sciatic notch only becomes more male in morphology between the ages of 6 and 15 years, which could be one of the reasons for the wrong assessments in the cases presented here. However, the Chi-square test found no significant differences among non-adults of different ages. This is slightly surprising, as besides the observation on the sciatic notch morphology [56], studies show that the development of the skeleton in the prenatal stage and in the first year of life is influenced by the production of near-adult levels of sex hormones [39,57,58,59]. However, populational differences, hormonal shifts, and inter- and intra-observer repeatability should also be acknowledged. Thus, the limitations of morphological methods for sex assessment of non-adults should always be considered.

Comparisons of morphologically assessed and genetically determined sex show that relatively straightforward skull- and pelvic-based morphological methodologies for sex assessment of non-adult and adult individuals work well in the Slovenian population from a modern-era cemetery of Ljubljana–Polje, and that pelvic-based assessment is more reliable when compared to the skull-based assessment. However, the assessment accuracy is not 100%. Thus, the sex assessment of individuals with genetically undetermined sex remains tentative. On the other hand, genetically determined sex solved the problem of 22 individuals with morphologically undetermined sex.

## 5. Conclusions

The study presented here demonstrates how morphological and molecular methodologies can go hand in hand when building a biological profile of an individual and/or studying the demography, social structure, cultural habits, etc., of a population. On their own, each methodology found some individuals with undetermined sex, while together, sex of all the individuals was provided. Though the accuracy of the morphological methods is not as good as the accuracy of genetic methods, especially for non-adults and adults without pelvic bones, usable information can be obtained. Genetically determined sex also confirmed that the chosen morphological methods for sex assessment are appropriate for the studied population. However, caution is needed with non-adults and adults without a pelvis.

## Figures and Tables

**Table 1 genes-14-01561-t001:** Age groups based on the age of the individuals.

Age (Years)	Age Group
<1	1
1–6	2
7–12	3
13–17	4
≥18	5

**Table 2 genes-14-01561-t002:** Sample information with sex assessment and sex determination results.

Sample	(Non) Adult	Age	Skull		Pelvis			Anthropological Sex	Genetic Sex
			Mandible	Skull	Ilium	Subpubic region	Greater Sciatic Notch		
1	adult	mature adult		male		male	male	male	male
2	adult	mature adult		NP		NP	male	male	male
3	adult	older adult		male		NP	NP	male	male
4	adult	mature adult		female		NP	female	female	female
5	non-adult	0–3 m	female		female			female	female
6	non-adult	1–2 y	female		NP			female	female
7	non-adult	38–40 w in utero	NP		NP			undetermined	male
8	non-adult	0–6 m	NP		NP			undetermined	undetermined
9	adult	young adult		NP		NP	female	female	female
10	non-adult	6–7 y	male		female			female	male
11	adult	mature adult		male		NP	male	male	male
12	non-adult	16.5 ± 1 y	NP		male			male	male
13	adult	young adult		NP		NP	female	female	undetermined
14	adult	older adult		female		NP	male	male	male
15	adult	adult		female		NP	female	female	female
16	adult	adult		male		NP	NP	male	male
17	adult	mature adult		female		female	female	female	female
18	non-adult	3.5 ± 1 y	male		female			female	male
19	adult	young adult		female		female	female	female	female
20	adult	mature or older adult		female		NP	NP	female	female
21	adult	older adult		female		NP	female	female	female
22	adult	mature adult		NP		female	female	female	female
23	adult	adult		male		NP	NP	male	male
24	non-adult	11.5 ± 1 y	NP		NP			undetermined	male
25	adult	young adult		female		NP	NP	female	female
26	non-adult	6.5 ± 1 y	female		female			female	male
27	non-adult	6–12 y	NP		NP			undetermined	male
28	adult	mature or older adult		male		NP	NP	male	undetermined
29	adult	older adult		male		NP	NP	male	female
30	adult	mature or older adult		male		NP	NP	male	male
31	non-adult	5.5–6.5 y	NP		female			female	female
32	non-adult	11.5–12.5 y	male		NP			male	female
33	non-adult	11.5 ± 1 y	female		female			female	female
34	adult	older adult		female		NP	NP	female	female
35	adult	older adult		male		NP	male	male	male
36	adult	adult		female		NP	NP	female	female
37	non-adult	0 ± 3 m	male		male			male	male
38	non-adult	38–40 w in utero	NP		NP			undetermined	female
39	non-adult	10.5–18 m	female		female			female	undetermined
40	non-adult	4.5 ± 3 m	female		female			female	male
41	non-adult	6.5–7.5 y	female		female			female	female
42	non-adult	10.5 ± 3 m	female		female			female	female
43	non-adult	38–40 w in utero	NP		NP			undetermined	male
44	non-adult	11–13 y	female		female			female	female
45	non-adult	12.5 ± 1 y	female		female			female	undetermined
46	adult	mature adult		female		NP	female	female	female
47	non-adult	11.5 ± 1 y	NP		NP			undetermined	male
48	adult	mature adult		male		male	male	male	male
49	non-adult	5–6 y	female		female			female	female
50	non-adult	3.5–4.5 y	female		female			female	female
51	non-adult	38–40 w in utero	female		female			female	female
52	non-adult	1.5–2.5 y	male		male			male	male
53	adult	young adult		male		male	male	male	male
54	adult	older adult		female		NP	female	female	female
55	non-adult	38–40 w in utero	NP		NP			undetermined	undetermined
56	non-adult	1.5–2.5 y	female		female			female	female
57	non-adult	2–3 y	female		NP			female	female
58	non-adult	4.5–5.5 y	NP		NP			undetermined	male
59	non-adult	10.5 m–1.5 y	NP		NP			undetermined	female
60	non-adult	7.5 ± 1 y	NP		NP			undetermined	female
61	non-adult	9–10 y	NP		male			male	female
62	non-adult	1–6 y	NP		NP			undetermined	female
63	non-adult	4.5–7.5 m	female		female			female	female
64	non-adult	10.5–12 y	female		female			female	female
65	non-adult	3.5–4.5 y	NP		female			male	female
66	non-adult	11.5–12.5 y	male		NP			male	male
67	adult	older adult		male		NP	male	male	male
68	non-adult	1–2 y	NP		NP			undetermined	male
69	non-adult	1–2 y	male		NP			male	female
70	non-adult	1.5–2.5 y	NP		male			male	male
71	non-adult	5.5–6.5 y	male		NP			male	male
72	non-adult	0–3 m	NP		NP			undetermined	male
73	non-adult	38 w in utero	NP		NP			undetermined	female
74	non-adult	5 ± 1 y	NP		female			female	female
75	non-adult	38–40 w in utero	female		female			female	female
76	non-adult	0–3 m	female		female			female	female
77	non-adult	10.5 m–1.5 y	female		NP			female	female
78	adult	mature adult		female		NP	female	female	female
79	non-adult	38–40 w in utero	NP		female			female	female
80	non-adult	4.5 ± 3 m	male		male			male	male
81	non-adult	1.5–2.5 y	female		female			female	male
82	non-adult	7.5 ± 1 y	NP		NP			undetermined	female
83	non-adult	6.5–7.5 y	female		female			female	male
84	non-adult	1.5 ± 1 y	female		female			female	female
85	non-adult	7.5 ± 1 y	male		male			male	female
86	non-adult	0–3 y	NP		NP			undetermined	male
87	non-adult	1–6 y	NP		female			female	female
88	non-adult	5.5 ± 1 y	male		male			male	male
89	non-adult	1–2 y	NP		female			female	undetermined
90	adult	older adult		male		NP	female	female	female
91	non-adult	38–40 w in utero	NP		male			male	male
92	non-adult	30–38 w in utero	female		NP			female	male
93	non-adult	5.5–6.5 y	female		female			female	female
94	non-adult	1–3 y	male		male			male	male
95	non-adult	0 ± 3 m	NP		male			male	male
96	non-adult	40 w in utero	NP		NP			undetermined	male
97	non-adult	6.5–7.5 y	female		NP			female	male
98	non-adult	4.5–7.5 m	NP		NP			undetermined	female
99	non-adult	7.5–10.5 m	NP		NP			undetermined	male
100	non-adult	10.5 m–1.5 y	female		NP			female	female
101	non-adult	6.5–7.5 y	male		female			female	female
102	non-adult	40 w in utero	NP		female			female	male
103	non-adult	9.5 ± 1 y	female		female			female	female
104	non-adult	5.5–6.5 y	NP		female			female	male
105	non-adult	7.5–8.5 y	female		NP			female	female
106	non-adult	32–34 w in utero	NP		NP			undetermined	male
107	adult	older adult		male		NP	NP	male	male
108	non-adult	30–34 w in utero	NP		NP			undetermined	female
109	non-adult	34–36 w in utero	NP		NP			undetermined	male
110	non-adult	4.5–7.5 m	female		NP			female	female
111	non-adult	34–38 w in utero	female		female			female	male
112	non-adult	1.5–2.5 y	female		female			female	female
113	non-adult	36–38 w in utero	NP		NP			undetermined	female
114	non-adult	38–40 w in utero	NP		female			female	female
115	non-adult	1.5–2.5 y	male		NP			male	male
116	adult	mature adult		NP		NP	female	female	female

## Data Availability

The datasets generated during and/or analyzed during the current study are available from the corresponding author on reasonable request.

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
