# Peer review of "Comparison of Morphological Sex Assessment and Genetic Sex Determination on Adult and Sub-Adult 17th–19th Century Skeletal Remains"

_genes, 2023, doi:10.3390/genes14081561_

Round 1
Reviewer 1 Report
In the manuscript " Comparison of morphological sex assessment and genetic sex determination on 17th – 19th century skeletal remains”, the authors presented a study comparing the sex of adult individuals and non, DNA determined and morphologically assessed, buried in a Early Modern (17th to late 19th century) cemetery in Ljubljana, Slovenia.
It is an article where the data are well presented and statistically valid.
The study underlines the importance of combining anthropological and molecular methods in order to accurately determine the sex of archaeological skeletal remains, especially in non-adult individuals or when there is poor conservation of the finds.
The research is not innovative, it agrees and reinforces previous well-established research that has already highlighted these aspects.
A significant finding of the study is that it has evaluated the macroscopic methods most commonly used in determining sex on the Slovenian population (from the 17th to the end of the 19th century).
While considering the study not original with a limited contribution to the ancient DNA scientific community, the authors have adequately directed the research therefore, considering the robustness of the data presented, the methodological procedures used and the analytical insights, it is recommended that it be published in Genes.
Author Response
Thank you for taking the time and reviewing our article. From the comments, we concluded that there is nothing major we can change in the manuscript. The report does state that certain parts could be improved, but the report is not specific. So we do not know what exactly is the problem.
Thank you again for the review.
Reviewer 2 Report
Dear Editor of Genes, and Authors,
I have read the paper titled « Comparison of morphological sex assessment and genetic sex determination on 17th – 19th century skeletal remain» with interest, finding it both theoretically and empirically consistent. It features a very interesting and useful comparison between morphological and molecular methods for the estimation of sex of human skeletal remains. In general, the paper is very good, with only some modifications needed including a rephrasing of “individuals with ambiguous sex characteristics” and “physical (biological is better) anthropologists”. Also, 71% accuracy in the non-adults sample for the morphological sex is not a good accuracy, especially in forensic cases, thus the sentence must be toned down.
My best regards.
Author Response
Thank you for taking the time and reviewing the article. We agree that the morphological sex assessment on non-adults is not good. We changed the sentence to emphasize that.